# Pharmacokinetic Analysis of Omomyc Shows Lasting Structural Integrity and Long Terminal Half-Life in Tumor Tissue

**DOI:** 10.3390/cancers15030826

**Published:** 2023-01-29

**Authors:** Marie-Eve Beaulieu, Sandra Martínez-Martín, Jastrinjan Kaur, Virginia Castillo Cano, Daniel Massó-Vallés, Laia Foradada Felip, Sergio López-Estévez, Erika Serrano del Pozo, Hugo Thabussot, Laura Soucek

**Affiliations:** 1Peptomyc S.L., Vall d’Hebron Barcelona Hospital Campus, 08035 Barcelona, Spain; 2Preclinical & Translational Research Program, Vall d’Hebron Institute of Oncology (VHIO), Vall d’Hebron Barcelona Hospital Campus, 08035 Barcelona, Spain; 3Department of Biochemistry and Molecular Biology, Universitat Autònoma de Barcelona, 08193 Bellaterra, Spain; 4Institució Catalana de Recerca i Estudis Avançats (ICREA), 08010 Barcelona, Spain

**Keywords:** protein therapeutics, MYC, Omomyc, mass spectrometry, LC-PRM

## Abstract

**Simple Summary:**

The maintenance of the structural integrity of therapeutic proteins in the target tissue is crucial to their proper function. In this study, we aimed to assess the in vivo stability of the therapeutic protein Omomyc in blood serum and tumor tissue in a xenograft mouse model of colorectal cancer. As Omomyc represents a new clinical modality to target MYC, a most wanted target found deregulated in the majority of human cancers, our findings provide grounds to support the administration regimen in solid tumors. Moreover, we show that this method could apply to paraffin-embedded clinical biopsies for direct protein detection in patient samples.

**Abstract:**

MYC is an oncoprotein causally involved in the majority of human cancers and a most wanted target for cancer treatment. Omomyc is the best-characterized MYC dominant negative to date. In the last years, it has been developed into a therapeutic miniprotein for solid tumor treatment and recently reached clinical stage. However, since the in vivo stability of therapeutic proteins, especially within the tumor vicinity, can be affected by proteolytic degradation, the perception of Omomyc as a valid therapeutic agent has been often questioned. In this study, we used a mass spectrometry approach to evaluate the stability of Omomyc in tumor biopsies from murine xenografts following its intravenous administration. Our data strongly support that the integrity of the functional domains of Omomyc (DNA binding and dimerization region) remains preserved in the tumor tissue for at least 72 hours following administration and that the protein shows superior pharmacokinetics in the tumor compartment compared with blood serum.

## 1. Introduction

MYC proteins are a family of transcription factors (c-, N-, and L-MYC), which coordinate transcriptional programs that allow cells to efficiently proliferate. Indeed, MYC provides building blocks for cell growth through control of metabolism and proliferation, but has also been described as being able to foster immune evasion in physiological and pathological conditions [1]. In order to function, MYC must heterodimerize with its obligate partner MAX and bind DNA via its basic, helix-loop-helix, leucine zipper domain (B-HLH-LZ). Such binding to MAX and to genomic target locations containing the E-box (enhancer box) sequence enables MYC to recruit multiple transcriptional cofactors and regulators of chromatin structure through its large and unstructured transactivating domain (TAD) [2].

MYC is typically found deregulated in human cancers, for instance, following gene amplification or alterations in upstream pathways controlling MYC protein expression, turnover, or stability [3]. Although it has been termed a “most wanted target for cancer treatment”, MYC has remained considered undruggable for a long time. Some of the reasons underlying such presumed undruggability include (1) the intrinsically disordered nature of the protein, which makes it challenging to design small molecules with sufficient affinity and selectivity for the target; (2) its localization within the nuclear compartment, often impermeable to drugs; and (3) the perceived risk of causing dramatic side effects in normal tissues, where MYC is considered essential to normal development and tissue regeneration [4].

Omomyc is a 91-amino-acids-long polypeptide that derives from the B-HLH-LZ domain of human MYC, containing four amino acid substitutions [5]. Because of these mutations, Omomyc is capable of forming homodimers and heterodimers with MYC and with MAX [5,6,7,8,9]. Such dimers blunt MYC activity by preventing its access to target DNA sequences (E-boxes) and replacing it by inactive complexes. Omomyc has served as a key research tool to increase our understanding of MYC biology in different experimental models, including normal and cancer cells and various mouse models of cancer. Importantly, the transgenic expression of Omomyc in vivo was employed to model for the first time systemic MYC inhibition, demonstrating a dramatic therapeutic impact and the viable therapeutic window of inhibiting MYC in cancer. Indeed, Omomyc expression led to tumor regression while preserving normal organ function [6,10,11]. Omomyc’s mode of action has been validated by different groups as the best MYC inhibitory strategy tested so far [6,9,12]. 

We previously uncovered the unexpected cell-penetrating properties of the recombinant Omomyc protein and established the preclinical proof of concept of using this miniprotein as a directly delivered biologic agent for cancer treatment [8]. We found that Omomyc penetrates cells mainly via multiple adenosine triphosphate (ATP)–dependent endocytic pathways, including, but not restricted to, macropinocytosis and clathrin- and caveolin-dependent endocytosis. This internalization is mediated by its N-terminal basic region, as the mutation of positively charged residues therein impairs cell penetration. Once internalized, the Omomyc miniprotein crosses the nuclear membrane and localizes significantly in the nuclei, where it displaces MYC from its genomic locations, causing a specific shutdown of MYC transcriptional signature and effectively stopping cell cycle progression [6]. These findings were independently confirmed by other groups as well [13]. In addition, we showed that upon intranasal administration to a KRAS^G12D^-mutated mouse model of lung adenocarcinoma, the Omomyc miniprotein colocalizes with lung tumors and blocks MYC-driven transcription and proliferation in vivo, where it also induces apoptosis and recruitment of intratumoral immune infiltrates, overall significantly abrogating tumor progression and reducing tumor grading [8]. Importantly, a similar effect was achieved by the intravenous (i.v.) administration of the Omomyc miniprotein, which allowed for systemic delivery and resulted in a wide therapeutic window in vivo, as well as synergistic efficacy when combined with standard chemotherapy in an erlotinib-resistant, EGFR-, P53-, and PI3K-mutated lung adenocarcinoma xenograft model [8]. More recently, the efficacy of i.v.-delivered Omomyc as both a single agent and in combination with paclitaxel also showed promising therapeutic profiles against metastasis in preclinical triple-negative breast cancer models [14].

However, while the blood pharmacokinetic (PK) profile and therapeutic window of Omomyc in murine models are well described by the available data [8], its stability within the tumor cells and in their vicinity after i.v. administration remained somewhat questioned [13]. In addition to chemical modifications, protein biotherapeutics can undergo proteolytic alterations and cleavage in vivo, which may lead to their functional inactivation and potentially affect the active drug pharmacokinetic profile. In fact, in cancer, different families of proteases are frequently dysregulated, where they are associated with various stages of disease development encompassing tumor growth, invasion, and metastatic spread and covering extracellular matrix remodeling, epithelial-to-mesenchymal transition, immune system evasion, and resistance to apoptosis signals, among others [15]. Therefore, the reliable measurement of functionally active levels of a drug at the site of interest is critical.

Recent advances in label-free mass spectrometry techniques have made them increasingly applicable to clinical proteomics, where they enable a wide range of tissue biopsy studies [16]. Here, we used an LC–MS/MS assay to assess the structural integrity of the Omomyc miniprotein in preclinical tumor biopsies. The quantification of functional Omomyc in such biopsies revealed that rapidly (2 h) after i.v. administration, higher concentrations of the drug are reached in the tumors compared with serum and persist there with at least one order of magnitude higher concentration than in serum 72 h after administration.

Such information is especially relevant considering that Omomyc is now in a clinical trial (NCT04808362) [17], where the PK data, as is customary in clinical practice, are being collected by serum sample analysis [11]. The methodology presented here may provide a model to more accurately extrapolate drug tumor concentrations from serum quantifications.

## 2. Materials and Methods

### 2.1. Production of Serum and Tumor Samples

DLD-1 colorectal cancer cells were inoculated subcutaneously into the flank of 8-week-old BALB/cOlaHsd-Foxn1nu mice (5 × 10^6^/100 µL in phosphate-buffered saline), and tumors were allowed to grow up to 430 mm^3^. The protocol was approved by the Ethical Committee for the Use of Experimental Animals (CEEA) at the Vall d’Hebron Institut de Recerca (VHIR) and by the Direcció de Polítiques Ambientals i Medi Natural, Departament de Territori i Sostenibilitat, Generalitat de Catalunya (code: 11354). Mice were then randomized (*n* = 5 per group) and treated i.v. with 50 mg/kg Omomyc or vehicle and euthanized after 5 min, 2 h, 24 h, 48 h, or 72 h following treatment. Serum was harvested and tumor biopsies were collected, cut into ~3 mm^3^ pieces, and split into flash-frozen (FF) and formalin-fixed paraffin-embedded (FFPE) samples. The FF tissue samples from treated mice were processed and analyzed in triplicate in order to answer whether the distribution of Omomyc would be homogeneous throughout the whole tumor.

### 2.2. Sample Preparation for Mass Spectrometry

All solvents were high pressure liquid Chromatography (HPLC) grade from Sigma-Aldrich (Darmstadt, Germany), and all chemicals were obtained from Sigma-Aldrich where not stated otherwise. The Omomyc miniprotein and vehicle-treated tumor samples were prepared for mass spectrometry using Biognosys’s (Schlieren, Switzerland) optimized protocol. The FFPE samples were deparaffinized, homogenized, and protein-denatured; the FF tissue samples were homogenized and protein-denatured. The samples were denatured in Biognosys’s denature buffer, reduced in Biognosys’s reduction solution for 60 min at 37 °C, and alkylated using Biognosys’s alkylation solution for 30 min at room temperature in the dark. Subsequently, the digestion to peptides was carried out using trypsin (Promega, 1:50 protease: total protein) and Lys-C (Fujifilm Wako Chemicals, 1:200 protease: total protein) per sample overnight at 37 °C. The resulting Omomyc peptides were purified on C18 BioPureSPN MIDI spin columns (Nest Group), while the tissue peptides were purified on a C18 Oasis HLB μElution Plate 30 μM plate (Waters) according to the manufacturer’s instructions and dried down using a SpeedVac system. Peptides were resuspended in LC solvent A (1% acetonitrile, 0.1% formic acid (FA)) and spiked with Biognosys iRT kit calibration peptides. Peptide concentrations were determined using a UV–VIS spectrometer (SPECTROstar Nano, BMG Labtech) for Omomyc and using μBCA (Thermo Scientific^TM^ Pierce^TM^, Basel, Switzerland) for the tissue samples. For Omomyc quantification, a set of 4 peptides representing Omomyc protein (ATAYILSVQAETQK, LISEIDLLR, DQIPELENNEK, THNVLER) and 1 peptide representing human MYC (SSDTEENVK) and their corresponding stable isotope synthetic (SIS) reference peptides were generated and spiked at known concentrations (Vivitide, ±10% quantification precision, >95% purity).

### 2.3. Shotgun LC–MS/MS for Spectral Library Generation

For the DDA LC–MS/MS measurements, 1 μg of peptides per sample was injected into an in-house packed reverse phase column (PicoFrit emitter with 75 μm inner diameter, 60 cm length, and 10 μm tip from New Objective, packed with 1.7 μm Charged Surface Hybrid C18 particles from Waters) on a Thermo Scientific^TM^ EASY-nLC^TM^ 1200 nanoliquid chromatography system connected to a Thermo Scientific^TM^ Orbitrap Fusion^TM^ mass spectrometer equipped with a Nanospray Flex^TM^ ion source. LC solvents were (A) 1% acetonitrile in water with 0.1% FA (B) 20% water in acetonitrile with 0.1% FA. The nonlinear LC gradient was 1–59% solvent B in 55 min, followed by 59–90% B in 10 s, 90% B for 8 min, 90–1% B in 10 s, and 1% B for 5 min at 60 °C, and a flow rate of 250 nl/min. A modified top speed method (3 s cycle time) from Hebert et al. was used [18]. The mass spectrometric data were analyzed using Biognosys’s search engine SpectroMine^TM^, and the false discovery rate on peptide and protein levels was set to 1%. A mouse UniProt fasta database (*Mus musculus*, 2021-07-01) and Omomyc protein sequence were used for the search engine, allowing for 2 missed cleavages and variable modifications (N-terminal acetylation, methionine oxidation).

### 2.4. LC-PRM Mass Spectrometry Acquisition and Data Analysis

For the LC-PRM measurements, 1 μg of peptides per sample were injected into an in-house packed reverse-phase column on a Thermo Scientific^TM^ EASY-nLC^TM^ 1200 nanoliquid chromatography system connected to a Thermo Scientific Q Exactive HF mass spectrometer equipped with a standard nano-electrospray source. LC solvents were (A) water with 0.1% FA and (B) 80% acetonitrile in water with 0.1% FA. The LC gradient was 1–59% solvent B in 55 min in nonlinear increments, followed by 90% B for 7.5 min (total gradient length was 67 min). A standard DIA mode run was acquired for retention time-based scheduling using Biognosys’s high-precision normalized iRT concept [19]. Signal processing and data analysis were carried out using the SpectroDive^TM^ 10 (Biognosys) software for multiplexed MRM/PRM data analysis based on mProphet [20]. A Q-value filter of 1% was applied. A data matrix of absolute quantities in fmol/μg peptides was generated (Q-value filter of 0.01 was applied) using single-point calibration derived using the following formula: *Target/Reference × SIS amount on column (fmol).*

### 2.5. Statistical Analysis

Differences in the quantification of Omomyc between the FFPE, FF, and serum samples were determined using a two-way ANOVA and by applying Tukey’s multiple comparison correction. The *p*-values <0.05, <0.01, <0.001, and <0.0001 were represented with *, **, ***, and ****, respectively, and were considered significant. Statistical analyses were performed using GraphPad Prism 8 (Dotmatics, Boston, MA, USA).

## 3. Results

We first aimed at selecting representative peptides to monitor Omomyc and the total protein content of the samples during targeted LC-PRM mass spectrometry experiments. To do so, we used shotgun LC-MS/MS to generate a peptide inventory from trypsin-digested Omomyc and vehicle-treated tumor samples. The sequence coverage of the 26 peptides detected from the digested Omomyc miniprotein reached 91%, encompassing the entire B-HLH-LZ domain (Figure 1 and Appendix A). Of those peptides, we selected a subset of four (THNVLER, DQIPELENNEK, ATAYILSVQAETQK, and LISEIDLLR) to be used for targeted quantification, which includes two proteotypic peptides for Omomyc (ATAYILSVQAETQK and LISEIDLLR). The selection criteria applied included (1) a length of 7–20 amino acids, (2) no extreme hydrophilicity/hydrophobicity, (3) the highest response on the mass spectrometer, and (4) no missed cleavages. In addition, we defined one negative control peptide representing the human MYC protein (SSDTEENVK).

The total peptide inventory from the vehicle-treated tumor samples included 3114 proteins with an average of 6.9 peptides per protein, including 21,623 unique modified peptide sequences and 28,540 peptide ion variants. A set of 10 control peptides shared between human and mouse FFPE and FF samples, and displaying stable levels across an array of tissues, were selected to be monitored in a label-free mode in LC-PRM measurements. For serum samples, five mouse control proteins were monitored in label-free mode.

We then proceeded to quantify Omomyc from serum, FFPE, and FF tissue samples using LC-PRM. The four peptides representing Omomyc and one peptide representing MYC and the corresponding SIS reference peptides were monitored for absolute quantification (Figure 2 and Appendix A). Omomyc was successfully detected and quantified in all the Omomyc-treated samples (Appendix A). Interestingly, the kinetic profile of all four representative peptides indicates that Omomyc rapidly (5 min) distributes to the tumor tissue and that the drug tumor concentration remains superior to serum levels from 2 h onward. Moreover, tumoral Omomyc concentrations decay slowly, remaining practically stable between, at least, 48 and 72 h after intravenous administration (Table 1). As evidenced in the table, it seems that we consistently detect a higher percentage of Omomyc left in the FF tissue compared with FFPE with the peptides LISEIDLLR and ATAYILSVQAETQK. Importantly, the kinetic profiles of the four Omomyc reporter peptides appear equivalent, indicating that the structural integrity of the quantified Omomyc is maintained throughout. No Omomyc detection was observed in the vehicle-treated samples, while all five serum and nine tissue control proteins included in the measurements were detected in all samples (Appendix A). Their relative intensity profiles were overall similar across the different specimens and showed minor sample preparation and loading differences, accounted for by normalizing each sample to the average of all control proteins. The human MYC peptide was not detected in any of the analyzed samples. Of note, no other MYC-derived peptides from the large (54 kDa) N-terminal region of MYC could be detected either. Taken together, these results indicate that both THNVLER and DQIPELENNEK can be safely attributed to Omomyc. However, because we cannot fully exclude that values derived from these two shared peptides could be coming from human MYC to some extent, we consider the other two specific peptides (LISEIDLLR and ATAYILSVQAETQK) to be more reliable for future quantifications.

Interestingly, the three pieces of tumor (FF samples) showed low intersample variability (SD < 1 in most cases, never above 2) in the Omomyc quantification, suggesting that it likely distributes homogeneously in the subcutaneous tumors. Similarly, there were no statistically significant differences between the quantifications in FFPE and FF samples.

## 4. Discussion

Successfully blocking MYC function remains a holy grail for human cancer treatment. Indeed, MYC is found deregulated in the majority of cancers, where it controls the transcription of genes that altogether enable tumors to proliferate, thrive, and even resist to different therapies. However, the development of a clinically viable MYC inhibitor has been challenging, especially due to technical difficulties associated with the intrinsically disordered nature of the protein, so that, to date, there is no MYC inhibitor approved for clinical use yet. In this context, Omomyc is the first clinical-stage cell-penetrating peptide targeting MYC for cancer treatment.

The initial preclinical studies on the purified Omomyc miniprotein revealed its long terminal half-life in plasma (estimated to be >60 h in mice) [8,13]. However, the proportion of tumor distribution and, most importantly, the structural integrity of the drug at the target site remained unknown. Indeed, only indirect approaches, making use of either radioactively or fluorescently labelled Omomyc, have been used to tackle this question [8,13,14]. However, the use of such covalent labels binding to polar or charged residues may alter the intrinsic stability of the therapeutic protein of interest, or could reflect the kinetic behavior of only a portion of the functionally active molecule.

Here, we use a label-free, high-resolution mass spectrometry approach to assess for the first time the distribution of intact Omomyc to the tumor following i.v. administration to mice. Our approach relies on four reporter peptides that cover the entire dimerization and DNA-binding domain of Omomyc. Our data confirm that not only does the Omomyc miniprotein reach the tumor after 2 h following intravenous administration at tissue concentrations within the range of serum’s, but also that the tumor concentrations are in fact higher than serum’s, and persist there for at least 72 h. Such long-lasting residence in tissue is coherent with the behavior of Omomyc as a folded protein, which can mainly be found in dimeric forms, rather than a peptide, more prone to quick degradation [13]. Notably, the relatively low intersample variability observed across the FFPE and FF samples is of special relevance given that clinical material is often restricted to FFPE biopsies only. Hence, being able to accurately calculate the absolute levels of Omomyc (and other proteins) in such samples becomes an extremely useful tool for its quantification in human biopsies from clinical trial studies.

It is noteworthy that the lack of detection of the MYC peptide in our samples suggests that the assay sensitivity is likely lower than that of Western blot (WB) and immunohistochemistry (IHC), for instance, methods typically employed to quantify the endogenous or exogenous protein levels in cells and tissues, which often take advantage of significant signal amplification [21].

## 5. Conclusions

These results indicate for the first time that the Omomyc miniprotein behaves as a stable protein in tumor tissue and that the pharmacokinetic studies performed in blood (serum and plasma) samples, in preclinical models or in standard clinical practice, provide a likely underestimation of its distribution and persistence in the tumor compartment. In addition, the methodology described here, although not straightforward to be implemented in the normal routine of a clinical trial, could be used for more accurate quantification of other therapeutic miniproteins that might display different stability in tumor tissue compared with blood samples.

## Figures and Tables

**Figure 1 cancers-15-00826-f001:**
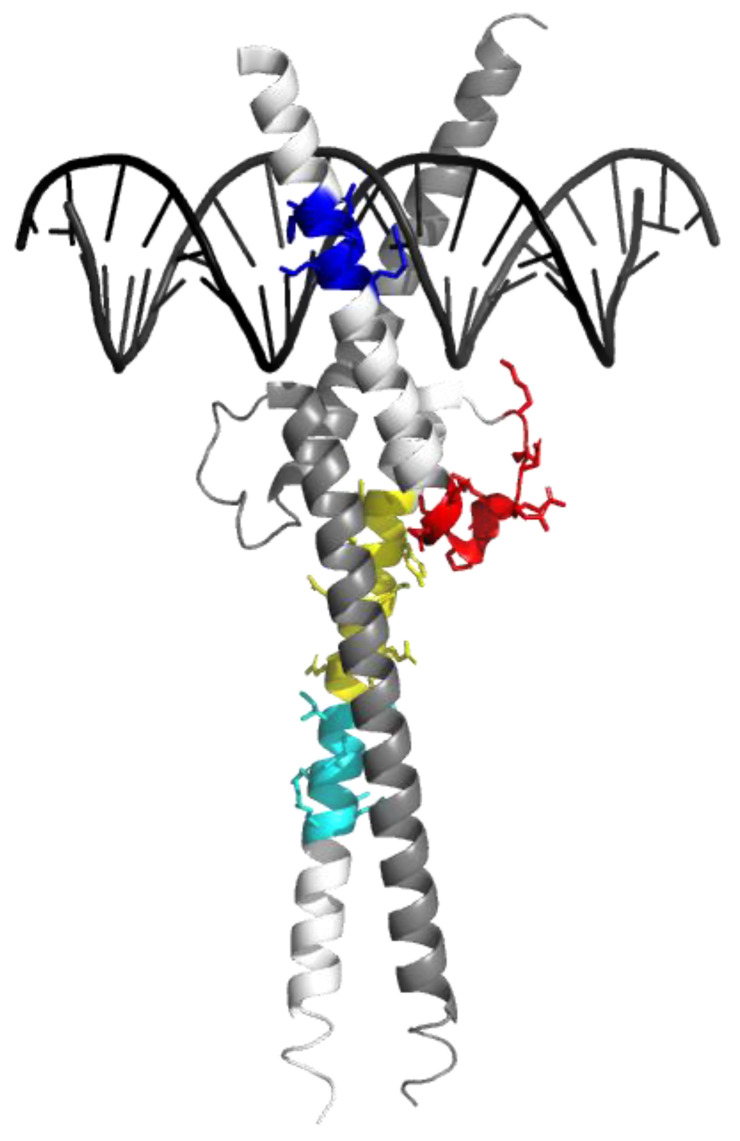
Reporter peptides are indicated on the tridimensional structure of Omomyc. Crystal structure of Omomyc (PDB ID 5I50.pdb) [9]. The reporter peptides used in this study are highlighted within the structure in blue (THNVLER), red (DQIPELENNEK), yellow (ATAYILSVQAETQK), and cyan (LISEIDLLR).

**Figure 2 cancers-15-00826-f002:**
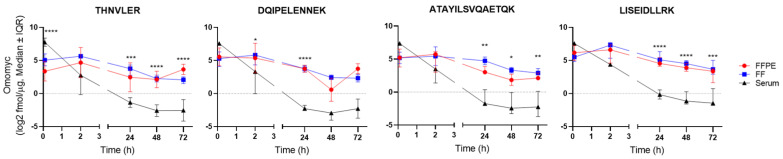
Absolute quantification of Omomyc in serum, FF, and FFPE tissue samples shows significant differences between the PK observed in tumor tissue compared with serum. Each representative peptide was quantified using single-point calibration (quantities derived from the ratios of endogenous to reference peak areas) and known spike-in amounts of SIS peptides. Data are shown as median ± interquartile range (IQR). * *p* < 0.05; ** *p* < 0.01; *** *p* < 0.001; **** *p* < 0.0001.

**Table 1 cancers-15-00826-t001:** Remaining percentage of Omomyc per sample relative to the earliest timepoint (5 min). Each absolute quantification of Omomyc in different samples was transformed into a percentage relative to the average of the 5 min timepoint per sample type and peptide. ND stands for not detected.

		FFPE Tissue			FF Tissue				Serum			
Sample ID	THNVLER	DQIPELENNEK	ATAYILSVQAETQK	LISEIDLLR	THNVLER	DQIPELENNEK	ATAYILSVQAETQK	LISEIDLLR	THNVLER	DQIPELENNEK	ATAYILSVQAETQK	LISEIDLLR
DLD-1 O5m.1	43.01	137.46	194.26	181.37	109.67	129.47	150.84	103.42	95.68	85.56	91.99	79.75
DLD-1 O5m.2	116.68	109.23	73.37	81.12	139.31	132.93	149.96	103.39	52.59	105.68	101.31	101.11
DLD-1 O5m.3	30.54	167.08	171.01	159.15	41.77	35.26	41.97	57.92	114.21	97.62	91.65	134.14
DLD-1 O5m.4	63.65	26.47	17.37	23.71	30.88	27.74	27.24	26.56	121.46	97.33	122.33	89.29
DLD-1 O5m.5	246.12	59.76	44.00	54.65	178.37	174.60	129.99	208.71	116.05	113.81	92.73	95.70
AVG	100.00	100.00	100.00	100.00	100.00	100.00	100.00	100.00	100.00	100.00	100.00	100.00
DLD-1 O2h.1	358.05	142.10	105.59	110.06	109.30	114.67	264.48	258.69	7.07	6.57	8.70	10.34
DLD-1 O2h.2	20.31	9.08	6.11	11.87	5.35	4.49	3.34	15.54	0.17	0.07	0.48	11.35
DLD-1 O2h.3	270.88	56.98	41.57	61.96	89.15	77.55	195.52	276.63	5.31	3.74	4.42	12.58
DLD-1 O2h.4	226.63	172.70	149.61	196.33	247.72	188.14	305.30	313.87	4.98	5.96	6.40	10.62
DLD-1 O2h.5	504.71	161.50	129.65	174.81	64.60	84.80	77.49	85.35	7.77	5.12	6.23	9.04
AVG	276.12	108.47	86.50	111.00	103.22	93.93	169.23	190.01	5.06	4.29	5.25	10.79
DLD-1 O24h.1	20.37	18.84	15.77	26.39	22.62	29.66	49.63	26.77	0.12	0.10	0.25	0.46
DLD-1 O24h.2	6.42	38.54	47.11	80.96	52.40	58.23	140.77	142.67	0.31	0.16	2.09	1.04
DLD-1 O24h.3	73.70	18.81	13.25	15.20	12.95	11.18	26.44	37.24	0.20	0.14	0.17	0.53
DLD-1 O24h.4	108.16	21.63	16.06	25.38	30.97	23.78	66.73	106.08	0.19	0.11	0.15	0.36
DLD-1 O24h.5	98.59	25.80	18.17	28.50	22.87	21.48	38.14	42.00	0.11	0.10	0.14	0.23
AVG	61.45	24.72	22.07	35.29	28.36	28.86	64.34	70.95	0.19	0.12	0.56	0.52
DLD-1 O48h.1	18.60	1.20	4.35	15.87	6.94	10.33	13.43	17.56	0.04	0.03	0.05	0.15
DLD-1 O48h.2	16.46	2.16	3.66	8.50	8.89	9.04	19.06	36.88	0.06	ND	0.08	0.19
DLD-1 O48h.3	28.22	2.55	8.24	17.48	8.56	10.28	13.50	16.56	0.07	ND	0.21	0.34
DLD-1 O48h.4	92.15	ND	20.66	39.83	12.32	10.49	31.25	59.31	0.10	0.07	0.10	0.23
DLD-1 O48h.5	40.46	7.51	7.06	16.81	12.86	15.89	26.39	50.64	0.14	0.07	1.33	1.12
AVG	39.18	3.35	8.79	19.70	9.92	11.21	20.73	36.19	0.08	0.06	0.35	0.41
DLD-1 O72h.1	169.93	42.40	15.67	15.90	10.46	12.72	35.94	50.85	0.05	ND	0.04	0.12
DLD-1 O72h.2	100.76	26.85	8.59	8.72	7.72	9.07	9.42	11.79	0.04	0.04	0.12	0.18
DLD-1 O72h.3	61.07	13.94	8.06	13.73	13.79	11.37	15.09	18.74	0.34	0.29	2.87	2.65
DLD-1 O72h.4	104.04	27.32	7.23	1.41	5.29	7.52	12.60	17.89	0.09	ND	0.13	0.27
DLD-1 O72h.5	59.59	17.95	13.78	11.53	10.31	12.29	20.13	27.98	0.04	ND	0.04	0.19
AVG	99.08	25.69	10.67	10.26	9.51	10.59	18.64	25.45	0.11	0.16	0.64	0.68

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
