# Peer review of "Pharmacokinetic Analysis of Omomyc Shows Lasting Structural Integrity and Long Terminal Half-Life in Tumor Tissue"

_cancers, 2023, doi:10.3390/cancers15030826_

Round 1

Reviewer 1 Report

The manuscript addresses a very specific and relevant question regarding MYC inhibition for cancer therapy. Through state of the art proteomics, the authors investigate the stability of the MYC inhibitor Omomyc in biopsies from subcutaneous tumors generated by the human colorectal cancer cell line DLD-1.

The work is well done and well presented. I feel that the authors should discuss or clarify several issues, listed below.

Small mistakes to be corrected:

- Figure 2: at the top of the the second plot y-axis, 0 should be substituted by 10. 

- In the first paragraph of the results, the authors have forgotten to include the third out of the four selection criteria applied.

Issues to be answered / discussed:

- it should be clarified whether there are special reasons for choosing DLD-1 cells as model.

- DLD-1 cells have high amounts of MYC, as reported by Dong Uk Kim et al., Oncology Letters, 17, 3589-3598. https://doi.org/10.3892/ol.2019.9996.

Therefore, the remark that the peptide representing human MYC is the negative control is unclear, and it is as well unclear why the MYC representing peptide was never detected in the proteomic analyses.

Probably the authors assume or find that there is no or undetectable MYC in DLD-1 cells. This ought to be shown.

- As the authors remark, two out of four peptides used for Omomyc quantification - LISEIDLRR and ATAYILSVQAETQK - are Omomyc specific. The other two are not, because they are non mutated in Omomyc vs human MYC bHLHZip region. So, they are present in human MYC. This might cause some confusion if MYC - as reported by Kim et al. - is significantly  expressed in DLD-1 cells.

- Tissue distribution and stability of Omomyc within cells at target sites was previously assessed through an Omomyc labelled with radioactive or fluorescent molecules (ref. 6 and 9). It was shown that Omomyc rapidly distributes to a variety of tissues, with a different stability in different tissues, and that it is bioactive. The authors should compare these data with those obtained through mass spectrometry. Basically, data reported in the manuscript confirm those obtained with labelled Omomyc. This may suggest that there is not a significant advantage in using the more complex and expensive mass spectrometry technique, not easily available to all scientists.

- Omomyc was shown to persist for about 70 h in plasma, with an initial rapid decay followed by a much slower decrease (ref. 6 and 9) . The plasma levels were much higher than the ones here reported for serum. Can the authors provide an explanation for this difference?

- In a lung adenocarcinoma model, it was evident the preferential tumor retention of Omomyc, compared to normal tissue (ref. 6). Does it occur here too? I wonder whether the authors have some evidence in this regard.

Author Response

The manuscript addresses a very specific and relevant question regarding MYC inhibition for cancer therapy. Through state of the art proteomics, the authors investigate the stability of the MYC inhibitor Omomyc in biopsies from subcutaneous tumors generated by the human colorectal cancer cell line DLD-1.

The work is well done and well presented. I feel that the authors should discuss or clarify several issues, listed below.

We thank the reviewer 1 for all the constructive input.

Small mistakes to be corrected:

- Figure 2: at the top of the the second plot y-axis, 0 should be substituted by 10. 

The axis has been corrected.

- In the first paragraph of the results, the authors have forgotten to include the third out of the four selection criteria applied.

We have corrected this typo by inserting the number 3) in the sentence where it was missing. The sentence now reads:

“The selection criteria applied included 1) a length of 7-20 amino acids, 2) no extreme hydrophilicity/hydrophobicity, 3) highest response on mass spectrometer, and 4) no missed cleavages. “

Issues to be answered / discussed:

- it should be clarified whether there are special reasons for choosing DLD-1 cells as model.

We chose the DLD-1 cells because they are fast growing in subcutaneous xenograft tumours and they display high MYC levels, so they represented a good model to explore whether MYC was above or below the limit of detection of the MS (see following point) and could somewhat interfere with Omomyc detection.

- DLD-1 cells have high amounts of MYC, as reported by Dong Uk Kim et al., Oncology Letters, 17, 3589-3598. https://doi.org/10.3892/ol.2019.9996.

Therefore, the remark that the peptide representing human MYC is the negative control is unclear, and it is as well unclear why the MYC representing peptide was never detected in the proteomic analyses.

Probably the authors assume or find that there is no or undetectable MYC in DLD-1 cells. This ought to be shown.

Given that no other peptide from the MYC protein is detected, including the human MYC control peptide used in this study (SSDTEENVK), we infer that MYC levels are below the limit of detection of MS (of note, other transcription factors, like MAX, are also undetectable). This is now indicated in the text, as specified in the next point.

In contrast, western blot and immunohistochemistry for MYC have been optimised for the target for several years and can take advantage of signal amplification (eg by chemiluminescence). Hence, they provide more sensitive detection methods compared to mass spectrometry.

One cannot exclude that sample preparation may influence which proteins are recovered successfully by MS, and it is possible that DNA-bound proteins are also less accessible to be enzymatically digested to yield positively charged peptides detectable by this technique.

- As the authors remark, two out of four peptides used for Omomyc quantification - LISEIDLRR and ATAYILSVQAETQK - are Omomyc specific. The other two are not, because they are non mutated in Omomyc vs human MYC bHLHZip region. So, they are present in human MYC. This might cause some confusion if MYC - as reported by Kim et al. - is significantly  expressed in DLD-1 cells.

As mentioned above, in addition to specifically measuring the positive control peptides for Omomyc, we also included a specific MYC reporter peptide in our spike tests (SSDTEENVK, not conserved in Omomyc), which was not detected. Moreover, we also looked for additional MYC peptides deriving from the large (~54 kDa) MYC N-terminal transactivating domain (TAD) and were not able to detect them in our samples.

We now included the following sentence in the discussion (line 236):

“Of note, the lack of detection of the MYC peptide in our samples suggests that the assay sensitivity is likely lower than western blot and immunohistochemistry typically employed to quantify endogenous transcription factors levels in cells. “

- Tissue distribution and stability of Omomyc within cells at target sites was previously assessed through an Omomyc labelled with radioactive or fluorescent molecules (ref. 6 and 9). It was shown that Omomyc rapidly distributes to a variety of tissues, with a different stability in different tissues, and that it is bioactive. The authors should compare these data with those obtained through mass spectrometry. Basically, data reported in the manuscript confirm those obtained with labelled Omomyc. This may suggest that there is not a significant advantage in using the more complex and expensive mass spectrometry technique, not easily available to all scientists.

The MS approach presented here study isn’t meant to replace the use of radioactively- or fluorescently-labelled protein detection, but rather to complement it. Indeed, our MS method is generally less accessible, more resource-intensive, and does require a more elaborate sample preparation compared to the other methods. However, it enables to capture an aspect of the drug stability that cannot be measured properly using neither the radioactive nor the fluorescent methods, as those rely on chemically modified drug.

Indeed, ref. 6 made use of a covalently-bound radioactive chelator attached to the C-terminal cysteine residue of the drug, while ref. 9 made use of covalently-bound fluorescent moiety attached to N-terminal amine-bearing residues (for instance arginine). Either modification could alter the protein stability and/or pharmacokinetics, especially the modifications to the N-terminal basic region, which target several positively charged residues, compared to the more specific C-terminal unique cysteine labelling employed for the radioactive labelling. Importantly, the N-terminal positively charged residues were shown to be critical for cellular internalization and for DNA binding (ref. 6 and 7), and thus their chemical modification is hypothesized to lead to poorer in vivo stability.

The clear advantage of the MS-based method we present here is to enable the use of unlabelled drug, a feature especially relevant in the context of its current clinical use.

A full pharmacokinetic profiling and quantification in all tissues was beyond the scope of our study, and unfortunately, the limited sample points we took do not enable a proper fitting nor real comparison with the PK profile from previous studies. Instead, we aimed at measuring drug integrity in the compartment of interest.

- Omomyc was shown to persist for about 70 h in plasma, with an initial rapid decay followed by a much slower decrease (ref. 6 and 9) . The plasma levels were much higher than the ones here reported for serum. Can the authors provide an explanation for this difference?

We have not compared side-to-side the recovery of Omomyc from serum vs plasma, but it is possible that the binding of Omomyc to serum proteins such as albumin yields differences between both types of samples preparations. A more in-depth methods characterization and validation study should be performed in order to properly address this question.

- In a lung adenocarcinoma model, it was evident the preferential tumor retention of Omomyc, compared to normal tissue (ref. 6). Does it occur here too? I wonder whether the authors have some evidence in this regard.

In the current study, we limited our analysis to tumour tissue vs blood pharmacokinetics and did not analyse other healthy tissues.

Reviewer 2 Report

This study by Soucek, Beaulieu and colleagues reports on the stability of Omomyc in tumor and serum samples from mice. This is important because Omomyc is currently in clinical trials as a fascinating new cancer therapeutic. The authors describe for the first time the detection of unlabeled Omomyc using a targeted proteomics approach that searches for reporter peptides. Because the authors were also able to determine Omomyc levels from FFPE samples, this method may be important for further evaluation of Omomyc's PK and PD behavior. The finding that Omomyc is much more abundant (i.e., probably more stable) in tumors than in serum is very interesting.

I think this study is well suited for publication, and I have only minor comments.

1. please add "immune-evasion" as a function of MYC to the sentence in lines 38/39, since that is what the cited review focuses on (or change the reference).
2. I would avoid the term "cutting-edge", it sounds like advertising.
3. please double check the references (maybe some have been mixed up?), e.g. ref. 7 is missing from the sentence in lines 46 - 48 and not all references in line 50 (e.g. ref. 4) are "independent".
4. please indicate whether the error bars in Figure 2 are SD or SEM.

Elmar Wolf

Author Response

This study by Soucek, Beaulieu and colleagues reports on the stability of Omomyc in tumor and serum samples from mice. This is important because Omomyc is currently in clinical trials as a fascinating new cancer therapeutic. The authors describe for the first time the detection of unlabeled Omomyc using a targeted proteomics approach that searches for reporter peptides. Because the authors were also able to determine Omomyc levels from FFPE samples, this method may be important for further evaluation of Omomyc's PK and PD behavior. The finding that Omomyc is much more abundant (i.e., probably more stable) in tumors than in serum is very interesting.

I think this study is well suited for publication, and I have only minor comments.

We thank the reviewer 2 for all the constructive input.

  1. please add "immune-evasion" as a function of MYC to the sentence in lines 38/39, since that is what the cited review focuses on (or change the reference).

The sentence now reads: “MYC proteins are a family of transcription factors (c-, N- and L-MYC), which provide building blocks for cell growth through control of metabolism and proliferation, in addition to fostering immune evasion”

  1. I would avoid the term "cutting-edge", it sounds like advertising.

We have removed the term from our manuscript.

  1. please double check the references (maybe some have been mixed up?), e.g. ref. 7 is missing from the sentence in lines 46 - 48 and not all references in line 50 (e.g. ref. 4) are "independent".

We have included ref 7. to lines 46-48, have removed the term “independently” to the sentence in line 50, and reviewed the references in the text.

  1. please indicate whether the error bars in Figure 2 are SD or SEM.

We have included the following sentence to the Figure 2 description: “Data are shown as median ± interquartile range (IQR).”

Reviewer 3 Report

Line 43.Upstream signalling pathways are also controlling Myc expression. Please, insert this information here.

Author Response

We have corrected the sentence, which now reads: “MYC is typically found deregulated in human cancers, for instance following gene amplification or alterations in upstream pathways controlling MYC protein expression, turnover or stability”.